# The Mitotic Cancer Target Polo-Like Kinase 1: Oncogene or Tumor Suppressor?

**DOI:** 10.3390/genes10030208

**Published:** 2019-03-11

**Authors:** Guillermo de Cárcer

**Affiliations:** Cell Cycle & Cancer Biomarkers Group, Cancer Biology Department, Instituto de Investigaciones Biomédicas “Alberto Sols” (IIBm), Consejo Superior de Investigaciones Científicas & Universidad Autónoma de Madrid, (CSIC-UAM), C/Arturo Duperier 4, 28029 Madrid, Spain; gdecarcer@iib.uam.es; Tel.: +34-91-585-4419

**Keywords:** polo-like kinase 1 (PLK1), mitosis, oncogene, tumor-suppressor, chromosome instability (CIN), aneuploidy, cancer

## Abstract

The master mitotic regulator, Polo-like kinase 1 (Plk1), is an essential gene for the correct execution of cell division. Plk1 has strong clinical relevance, as it is considered a bona fide cancer target, it is found overexpressed in a large collection of different cancer types and this tumoral overexpression often correlates with poor patient prognosis. All these data led the scientific community to historically consider Plk1 as an oncogene. Although there is a collection of scientific reports showing how Plk1 can contribute to tumor progression, recent data from different laboratories using mouse models, show that Plk1 can surprisingly play as a tumor suppressor. Therefore, the fact that Plk1 is an oncogene is now under debate. This review summarizes the proposed mechanisms by which Plk1 can play as an oncogene or as a tumor suppressor, and extrapolates this information to clinical features.

## 1. Introduction

The mitotic regulator gene Polo-like kinase 1 (Plk1) was identified 30 years ago in Drosophila melanogaster (named *polo*) [1,2]. *Polo* is closely related to the budding yeast gene *cdc5*, identified a decade before in the inspirational cell cycle studies done by L.H. Hartwell and colleagues [3,4]. Both, in *Drosophila* and yeast, Plk1 is an important factor for cell cycle progression, and more precisely, an essential gene for the mitotic process [2,5]. In 1994, the E. Nigg laboratory identified the mammalian orthologue (called Plk1) [6]. Since then, extensive studies have been performed depicting in detail all the PLK1 molecular functions during mitosis, cell cycle progression and even non-cell cycle related mechanisms (reviewed in [7]). Plk1 is the founding member of the polo-like kinase family, composed of five different genes, from Plk1 to Plk5 in higher eukaryotes [8]. The five Plk genes share a canonical kinase catalytic domain and a regulatory region composed by one or two Polo Boxes Domains (PBD), which confer the identity to the family and are involved in the binding to substrates (reviewed in [7,9]). PLK1 controls a wide variate of processes during the cell cycle progression, such as centrosome maturation, entry into mitosis, chromosome segregation and cytokinesis (reviewed in [10]). Accordingly, PLK1 has different subcellular localizations depending on the function to exert. From the cytoplasm and centrosomes during interphase, when the cells enter in mitosis, PLK1 concentrates at the spindle poles after the nuclear envelope breaks down (NEBD). When chromosomes in prophase condense, PLK1 shuttles to the kinetochores helping chromosomes to align in the metaphase plate. At the onset of anaphase, PLK1 localizes at the spindle mid-zone and later to the cytokinetic bridge in order to coordinate cytokinesis and cell abscission (reviewed in [9,10]). All these functions during mitosis, make Plk1 an essential gene for cell life. Inhibition of Plk1 function by RNA interference, small-molecule inhibitors, or genetic knock-outs result in aberrant mitotic progression. Cells entering in mitosis with no functional PLK1 are unable to establish a bipolar spindle and therefore cannot properly align chromosomes in the metaphase plate. This leads to the activation of the Spindle Assembly Checkpoint (SAC) and cells arrest in mitosis for longer times until they eventually die. This PLK1 inhibition phenotype is conserved from the yeast *cdc5*-null strains to Drosophila *polo*-null mutants [2,5] up to higher mammalian organisms such as mice where *plk1*-null embryos die at the morula stage due to a massive mitotic arrest [11].

Shortly after Plk1 was described in mammals, growing shreds of evidence showed that Plk1 is commonly upregulated in a wide variety of tumors, and its overexpression has been validated as a prognostic marker in a subset of these cancers [12,13,14,15,16,17]. In 1997, Plk1 was formally associated to malignant transformation mechanisms in the D.K. Ferris laboratory, where ectopic Plk1 overexpression experiments in murine fibroblast, were reflected in a higher proliferation index and caused oncogenic focus formation and xenograft tumoral growth [18]. All these features led the scientific community to consider Plk1 as an oncogene over decades (reviewed in [19,20,21]), despite its contribution to tumor development still being uncertain [22].

The fact that Plk1 is overexpressed in tumors, and that Plk1 inhibition is an efficient way to stop cancer cell proliferation positioned PLK1 as a promising cancer target. Consequently, pharmaceutical companies added PLK1 in their portfolios, and a growing collection of small inhibitors have entered several clinical trials in recent years [23], some of them reaching phase III stages and even recently receiving the “breakthrough designation therapy” by the FDA [24].

Unexpectedly, recent research done with genetically modified mouse models strongly suggests a possible tumor suppressor role for PLK1 [25,26,27,28], placing the oncogenic function of PLK1 under debate. The aim of this review is to put into context the latest evidence showing how Plk1 might play a role either as a tumor-prone or a tumor-suppressor gene and additionally, to correlate this information to a tumoral clinical outcome obtained from in-silico databases depending on the Plk1 expression levels.

## 2. Tumor-Prone Mechanisms of PLK1

Due to its essential role in cell proliferation, Plk1 is expressed in highly proliferative tissues or during proliferative events such as developing embryos, testis, thymus, spleen, etc. All of which are typical proliferating tissues where Plk1 expression levels are high [6]. Concomitantly, any tumoral tissue is susceptible to express high levels of Plk1 as a mere reflection of its higher proliferation rate. This fact keeps open the question of whether Plk1 overexpression in tumors is due to its direct contribution to the transformation status, or just reflects an increased proliferation rate. Even though the molecular mechanisms of Plk1 in tumor appearance and tumor progression are still under debate [22], recent studies have shed light on this issue, defining detailed molecular mechanisms on how Plk1 directly participates in the oncogenic signaling.

### 2.1. Plk1 Transcriptional Regulation Feedback Loops

Transcriptional control is a crucial feature of Plk1 regulation. Plk1 is transcriptionally regulated during the cell cycle progression mainly by the forkhead transcription factor, FOXM1. This leads to a peak of Plk1 expression at the G2/M phase of the cell cycle [29]. Besides, Plk1 includes binding motives for E2F transcription factors in its promoter region (CDE/CHR and CCAAT elements), and it has been described that E2Fs transcription factors can act either as activators or repressors of Plk1 gene expression [30,31,32]. Although E2F and FOXM1 are closely related to oncogenic processes, Plk1 regulation in tumors that are driven by any of these transcription factors might merely recapitulate the higher proliferation index within the tumor.

Plk1 gene expression is also controlled by other transcription factors such as p53 and Myc, both being intimately related to oncogenesis. A common mechanism to explain the potential role of Plk1 in cell transformation is the direct correlation between Plk1 and the transcription factor p53 (Figure 1a). The tumor suppressor p53 is induced by cellular stress as a safeguarding process, regulating the expression and/or repression of a large number of target genes leading to cell-cycle arrest and eventually apoptosis [33]. The TP53 gene is mutated in approximately 50% of all tumoral malignancies. In addition, many tumors that do not exhibit mutations on TP53 have inactivated the p53 function by other means, resulting in almost all tumoral malignancies having an impaired p53 function [34,35]. The Plk1 locus contains regulatory responsive elements for p53 in its promoter region at positions −2067 to −2016 [36]. As mentioned above, Plk1 harbors the canonical CDE/CHR cell cycle regulation binding sites [37] that can be directly modulated by p21, a p53 effector [29,32]. Plk1 expression is repressed by the binding of p53 and/or p21. Therefore, in tumors where the p53 function is impaired, one can expect that Plk1 levels are elevated, and this has been proposed as a possible mechanism by which p53 loss can contribute to the tumoral status. Interestingly, there is also evidence showing that Plk1 can modulate p53 function. Firstly, a direct interaction of Plk1 to p53 seems to be sufficient to inactivate p53 in lung tumoral cells [38]. Secondly, involving an indirect interaction, Plk1 can alter p53 stability by modulating the activity of MDM2 (a p53 stabilizer). MDM2 is phosphorylated in Ser260 residue by Plk1, and this increases the p53 destabilization [39]. Plk1 expression is also able to alter p53 phosphorylation status at Ser15, and this impacts on p53 stability due to impaired binding to the MDM2 [40]. In a similar trend, Plk1 is also able to phosphorylate two factors that regulate p53 activity. On one side, Topors (a topoisomerase I-binding protein) is phosphorylated at Ser718 by Plk1, leading to p53 ubiquitination and subsequent degradation [41]. On the other side, GTSE1 is phosphorylated at Ser435 by Plk1, which makes p53 excluded from the nucleus and therefore prone to be degraded [42]. All these data indicate that Plk1 and p53 are closely related in an inhibitory feedback loop that can induce oncogenic addiction to cells (Figure 1a). Tumoral cells that have lost p53 function will overexpress Plk1 and this can accelerate the cell cycle. Simultaneously, transformed cells that overexpress Plk1 can inhibit p53 function leading to a more aggressive oncogenic status.

Another transcription factor recently described to modulate Plk1 expression is oncogene Myc. Two independent reports have identified the Myc canonical binding E-box motif (CACGTG) at the Plk1 gene promoter region, located at -82 and -162 positions (respect to the transcription initiation site) [43,44]. In lymphoma and neuroblastoma cancer cells, Myc downregulation decreases the expression of Plk1. In accordance, Myc overexpression leads to higher Plk1 expression levels. More interestingly, and similarly to p53, Plk1 kinase activity can modulate Myc stability by two different proposed mechanisms. In one hand, in double-hit lymphoma (DHL) cells, Plk1 is able to activate the AKT-GSK3β kinase circuit, thus leading to an increase in Myc stability [44]. On the other hand, Plk1 phosphorylates the SCF^Fbw7^ ubiquitin ligase complex in the Fbw7 subunit. This phosphorylation induces Fbw7 self-ubiquitination and its subsequent proteasomal degradation, therefore increasing the accumulation of downstream targets such as Myc [43]. In summary, Plk1 and Myc constitute a positive feedback loop that explains how Plk1 can be directly implicated in oncogenic events, in this case by modulating the Myc oncogene stability (Figure 1b).

### 2.2. Plk1 Promotes the Inactivation of Tumor Suppressors

Plk1 can modulate protein abundance by phosphorylating phosphodegrons in a variety of substrates. This feature is essential for accomplishing several steps of mitosis. At the end of G2, and helping for the entry into mitosis, Plk1 phosphorylates Bora and Wee1 proteins allow their degradation to be mediated by the SCFβ^TRCP^ complex [45,46]. Also mediated by the SCFβ^TRCP^ complex, Plk1 can trigger the degradation of Emi1, leading to the APC/C-Cdc20 activation and subsequent exit from mitosis [47]. In a similar fashion, recent reports showed that Plk1 can participate in oncogenesis by promoting the degradation of proteins directly related to transformation events.

In triple-negative breast cancer (TNBC), the tumor suppressor REST protein levels are almost absent, although the REST gene is neither deleted nor mutated. Overexpression of Plk1 in TNBC cooperates with SCYL1 and TEX14 (the so-called “STP axis”) leading to reduced levels of the tumor suppressor REST. Plk1 phosphorylates REST at Ser1030 generating an SCFβ^TRCP^ phosphodegron, thus priming REST to protein degradation by the proteasome. Interestingly, the “STP axis” components are commonly amplified in TNBC. This feature determines the degradation of the tumor suppressor REST leading to TNBC primary tumor growth and metastatic expansion, and it also correlates with patient poor prognosis [48] (Figure 1c).

In a hepatocellular carcinoma, when originated by the infection of hepatitis B virus (HBV), Plk1 has been suggested to participate in the transformation event by modulating the degradation of two transcription repression factors, SUZ12 and ZNF198 [49] (Figure 1d). Degradation of SUZ12 and ZNF198 is critical for hepatocellular carcinoma appearance and progression, since both proteins are essential subunits of the polycomb repressive complex 2 (PRC2), and the LSD1-CoREST-HDAC1 repression complex respectively, with strong implications in cancer pathogenesis [50]. The HBV liver infection promotes the integration and expression of the viral protein HBx, an oncogene that allows the hepatocellular carcinoma appearance [51,52]. HBx has been described to directly activate Plk1 in the G2 phase of the cell cycle and this is reflected in an impaired DNA damage checkpoint and increase in polyploidy [53]. Plk1 increased activity leads to SUZ12 phosphorylation at residues Ser539, Ser541, and Ser546; and ZNF198 phosphorylation in Ser303, Ser305, and Ser309. In both cases, phosphorylation of these residues by Plk1 leads to phosphodegron activation and subsequent protein degradation. Interestingly, liver tumors with high levels of Plk1 often overexpress to the long, non-coding RNA HOTAIR. HOTAIR acts as a scaffold for ubiquitination events, by interacting with RNA binding E3 ubiquitin ligases [54]. In this trend, HOTAIR recruits SUZ12 and ZNF198 and synergizes with Plk1 phosphorylation allowing a more efficient degradation of both transcription repressors.

Plk1 can also modulate the “phosphatase and tensin homolog” (PTEN) during cell cycle progression (reviewed in [55]) (Figure 1e). PTEN is a well-known tumor suppressor that is mainly localized into the nucleus in primary cells. This nuclear localization is reduced in cancer cells, and the presence of PTEN at the cytoplasm, where it is less active, may help as an indicator of poor prognosis [56]. Plk1 is able to phosphorylate PTEN at Ser385 resulting in the inhibition of PTEN mono-ubiquitination, avoiding PTEN degradation, and helping to facilitate its accumulation at the cytoplasm, where it is less active [57]. In addition, PTEN is a direct negative regulator of the oncogenic kinase “phosphatidylinositol 3-kinase” (PI3K), which acts by antagonizing the PI3K effector AKT [58]. Agreeing to the negative regulation of PTEN mediated by the Plk1 in tumoral cells, Plk1 overexpression increases activation of the PI3K; whereas reduced levels of Plk1 decrease it. Therefore, the activation of the PI3K pathway, upon Plk1 overexpression, leads to an increased proliferation rate also accompanied by an elevated Warburg effect and higher tumorigenesis rates [57].

## 3. Plk1 Is Able to Play as a Tumor Suppressor

The original preliminary idea that reduced levels of Plk1 are also related to an oncogenic scenario, came in 2001 with the description of Plk1 mutations in certain tumoral cell lines. These mutations make Plk1 more unstable due to their incapacity to bind to the chaperone Hsp90. Hence, these tumoral cells supposedly have lower levels of Plk1 [59]. Unfortunately, these experiments were performed in cancer cell lines in vitro, and the described mutations have not been found in real cancer patients thus far. Indeed, Plk1 is rarely found mutated in tumors (about 1% of 74402 tumoral samples (as depicted from the cBioPortal database—http://www.cbioportal.org), and when mutated, this probably happens at late stages of the tumoral progression [60]. The low mutation rate in the Plk1 gene is most probably due to the fact that Plk1 is an essential cell proliferation gene, therefore cells cannot handle the Plk1 loss of function.

The first in vivo evidence showing that Plk1 can play as a tumor suppressor arose in 2008 with the first *Plk1* genetic modified mice strain generated by gene-trapping strategies [25]. Whereas, complete depletion of *Plk1* is incompatible with embryonic development, stopping mouse embryos at the morula stage [11,25], *Plk1* heterozygous mice are compatible with life and are tumor-prone. In the long term, *Plk1*(+/−) mice develop tumors with a higher incidence of lymphomas, also accompanied by lung carcinomas, squamous cell carcinomas, and sarcomas (Figure 2a). Splenocytes derived from these mice were tested for ploidy analysis, showing increased levels of aneuploidy, suggesting a potential mechanism by which reduced levels of PLK1 can be tumor prone. On the contrary, other *Plk1* depletion studies done in different *Plk1*-modified mice strains showed that reduced levels of PLK1 do not alter mouse viability in terms of cancer appearance [11,61,62]. Interestingly, these other mice strains do not show any significant alteration in cell proliferation or aneuploidy rates in derived mouse embryonic fibroblasts (MEF). There are also no alterations in blood cell populations, neither in the proliferation rates of different tissues, even in cases where PLK1 levels are reduced up to 90% [61].

In 2013, Plk1 was again proposed as a tumor suppressor in an elegant transcriptomic study done in human breast cancer cells by M. Beato and colleagues [26]. Authors proposed that Plk1 is able to directly modulate the estrogen receptor (ER) dependent gene transcription profile, by physically interacting with ER and therefore being recruited to the enhancer elements of ER-regulated genes. Surprisingly, when Plk1 activity is inhibited, the ER-dependent gene sets that were downregulated correlate with tumor-suppressive functions. Alongside, breast tumor patients that have an elevated expression signature of these Plk1-dependent gene set, positively correlate with a favorable clinical prognosis. In addition, authors suggest a possible mechanism by which PLK1 might phosphorylate the histone-lysine N-methyltransferase (MLL2) at Ser4822. MLL2 is a critical regulator of developmental genes in mice, and also interacts with ER, being a key player for the transcriptional activation of ER target genes. In this trend, and despite classical studies showing that Plk1 expression confers poor outcomes in breast cancer patients [63], there are reports showing the beneficial effects of Plk1 activity in breast cancer. For instance, Plk1 histological expression analysis showed that breast tumors with increased levels of PLK1 protein have a better prognosis when compared to the samples with very little or an absent PLK1 presence [64]. It is worth noting that the same study shows that Plk1 over-expression correlates with poor prognosis in p53 deficient breast cancer samples, in agreement with the p53-Plk1 feed-back loop as described above.

The potential role of Plk1 as a tumor suppressor in breast cancer has been recently validated using knock-in mouse models, in the M. Malumbres’ and R. Sotillo’s laboratories [27]. Overexpression of the human Plk1 cDNA, in a conditional inducible knock-in mouse (Plk1-KI), revealed that these mice can tolerate increased levels of Plk1, with no significant higher rates of tumor appearance when compared to control littermates that do not express Plk1. Thus, Plk1 seems not to play as an oncogene. Strikingly, when Plk1 inducible expression in mammary glands was combined with mice strains carrying either the *K-Ras* or *Her2* oncogenes, tumor incidence was dramatically reduced in both cases, up to 85% and 50%, respectively (Figure 2b). In addition, classical in vitro transformation assays done in immortal MEFs derived from these mice, revealed that Plk1 over-expression not only does not lead to cell transformation but also impedes transformation by oncogenes such as *K-Ras-V12*. Surprisingly, the same in vitro assays done in *p53*-null MEFs showed an efficient transformation suppression phenotype when Plk1 levels increased, leaving an open question about the p53-Plk1 connection in tumoral processes. The proposed mechanism by which Plk1 halts tumor appearance relays on the induction of severe polyploidy upon Plk1 overexpression. Cells with higher PLK1 levels cannot perform proper cytokinesis, failing in the final cytokinesis abscission event, what leads to elevated levels of polyploidy that are incompatible with cell proliferation. It is worth mentioning that another *Plk1*-KI mouse strain developed by X. Liu and co-workers also show that mice with higher *Plk1* do not develop tumors in the long latency. Interestingly, when these mice are subjected to ionizing radiation, then the *Plk1*-overexpressing cohort develop more tumors than the control non-*Plk1* expressing littermates [65]. The suggested underlying mechanism is that Plk1 over-expression reduces the efficacy of the DNA damage response signaling pathway. Thus, alterations due to the ionizing radiation cannot be efficiently repaired, leading to aneuploidy and tumor development.

The tumor suppression capacity of Plk1 has also been shown in other cancer models such as colorectal cancer [28] (Figure 2c). The K. Strebhardt’s laboratory has generated a mouse strain that combines a *Plk1* knock-down allele (by expressing an inducible shRNA against *Plk1* in the Rosa26 locus—(Plk1^iKD^) [61]) over the APC^min/+^ allele. The APC^min/+^ strain mimics the nonsense mutations present in the *adenomatous polyposis coli* gene (APC) which are commonly found in familial colorectal cancer patients. The APC^min/+^ mice spontaneously developed colorectal adenomatous polyps [66]. The resulting mouse strain (Plk1^iKD^; APC^min/+^), when induced to have lower levels of *Plk1*, showed an increased rate of colorectal adenomatous polyps appearance compared to the control littermates that harbor normal *Plk1* expression levels [28]. Similar data have been retrieved by using the PLK1 inhibitor Volasertib (BI6727) in the APC^min/+^ strain and also in human cells derived from colorectal tumors that express a truncated APC version. The proposed mechanism relays on a hampered Spindle Assembly Checkpoint (SAC) due to the truncated APC expression, which gets even further weakened when Plk1 is downregulated or inhibited. This SAC deregulation leads to aneuploidy and therefore increased tumor appearance.

As a summary, all these recent reports based on genetically modified mouse models, present convincing data on two main subjects. First, Plk1 over-expression does not play as an oncogene by itself, neither in vivo nor in vitro when tested in classical transformation assays. Second, Plk1 can play as a tumor suppressor when combined with certain oncogenes (such as K-Ras, Her2 or APC^min^), either by over-expression or by down-regulation strategies. Despite the different molecular and cellular mechanisms proposed, all them collide in the same idea of generating mitotic aberrations and the subsequent aneuploidy and/or polyploidy events that lead to chromosomal instability (CIN).

## 4. Clinical Consequences of Plk1 Upregulation

Already having evidence on how Plk1 can collaborate with the tumor process, and that it can also function as a tumor suppressor; it is of interest to test if this paradigm can be reflected in cancer patients, in terms of clinical outcome. As mentioned above, we can easily find Plk1 overexpression in a large variety of tumors, and this overexpression often confers poor prognosis to the patients [20,21,67]. On the other hand, in the studies showing that Plk1 play as a tumor suppressor, there is also data showing that Plk1 overexpression confers good prognosis to certain cancer subtypes. For example, analyzing the Cancer Genome Atlas (TGCA) database, we can observe that breast cancer samples that harbor genome doublings have a significant increase in Plk1 gene expression when compared to breast cancer samples that have a single genome copy. Of note, these breast cancer samples with higher Plk1 levels have better clinical prognosis according to the genome-doubling feature [27]. Moreover, we can recapitulate equivalent data from breast cancer patient samples by retrieving information from the Kaplan Meier plotter initiative (www.kmplot.com) [68], where gene expression data is correlated with patient clinical data like overall survival, relapse-free survival, therapy responses, etc. Plk1 mRNA over-expression can either confer poor or good prognosis in breast cancer, depending on the tumor subtype (Figure 3). For example, breast cancer patients with concomitant expression of Plk1 and the Estrogen Receptor (ER+) have a significantly worse relapse-free survival (RFS) index when compared to ER+ patients that have little or no Plk1 expression. On the contrary, Estrogen Receptor-negative (ER−) and Her2 amplified (Her2+) cancer patients, that have higher levels of Plk1, present a significantly improved RSF outcome. In case of the APC colorectal tumors, Strebhardt and colleagues showed similar data also using the TCGA database, identifying one hundred colon cancer patient samples that harbor the APC non-sense mutations for which PLK1 expression and clinical data is also available. In this case, patients with elevated Plk1 levels have a significantly better prognosis compared to patients with low Plk1 expression. Thus confirming the fact that Plk1 can also play as a tumor suppressor in APC related colon tumors. In summary, these data show that Plk1 can indeed play a double game, either cooperating with the tumor progression or being a tumor stopper depending on the tumoral genetic background.

Can we get a wider picture of this Plk1 dichotomy in a pan-cancer analysis? Again, we can make use of the Kaplan Meier plotter initiative (www.kmplot.com) [68], testing how Plk1 expression correlates with overall survival (OS) in a large collection of different cancer types. The most recent information implemented by the Kaplan Meier plotter initiative uses mRNA quantitative sequencing data in a collection of twenty different cancer types (Breast cancer, Cervical squamous cell carcinoma, Esophageal carcinoma, Head-neck squamous cell carcinoma, Kidney renal clear cell carcinoma, Kidney renal papillary cell carcinoma, Liver hepatocellular carcinoma, Lung adenocarcinoma, Lung squamous cell carcinoma, Ovarian cancer, Pancreatic ductal adenocarcinoma, Pheochromocytoma and Paraganglioma, Rectum adenocarcinoma, Sarcoma, Stomach adenocarcinoma, Testicular Germ Cell Tumor, Thymoma, Thyroid carcinoma, and Uterine corpus endometrial carcinoma). Correlation analysis of Plk1 expression with patient overall survival demonstrates that Plk1 overexpression leads to different outcomes depending on the tumor type (Figure 4). Plk1 expression leads to poor prognosis in lung, bladder, and kidney clear cell carcinomas patients, whereas in patients suffering from thymoma, lung squamous cell carcinoma or rectum adenocarcinoma, Plk1 higher levels seem to indicate a much better prognosis. There is also a group of tumors where Plk1 levels do not alter the patient prognosis (ovarian cancer, stomach carcinoma, and cervical squamous carcinoma). These data reveal that Plk1 expression levels can either specify poor or good prognosis to patients, hence showing that Plk1 can play as a tumor aggressiveness accelerator or as a tumor stopper in a wide variety of different tumors.

## 5. Discussion

Plk1 is a very well-known mitotic kinase, intimately related to cancer events since its description in mammals. In past decades, we have gathered a large number of reports showing that Plk1 is indeed overexpressed in many different cancer types and that this overexpression is crucial for the cancer progression [20,67]. Upon all this data, Plk1 is often considered an oncogene, and there are excellent reports showing how Plk1 can benefit the tumoral progression. As commented here, we have some evidence that shows that Plk1 can modulate tumor progression by controlling critical oncogenic transcription factors such as p53 and Myc. In addition, Plk1 can also modulate oncogenic signaling pathways such as the PI3K-MEKK, or dampening the function of well-known tumor suppressors such as PTEN or REST. It is worth highlighting two important features of these studies: Firstly, none of these mechanisms proposed are related to the mitotic functions of Plk1. It seems that Plk1 mitotic functions are not the drivers of tumorigenesis unless we consider aneuploidy as the mitotic-related oncogenic mechanism, as it happens to many other mitotic genes [69]. Secondly, all previous experimental data are performed in cells that are already tumoral, indicating that Plk1 can contribute to the tumoral status once already established, but does not necessarily demonstrate that Plk1 can play as a real oncogene, which means driving transformation by itself. Thus, experiments that demonstrate whether these Plk1 dependent mechanisms are sufficient to induce cell transformation are still pending to be performed.

Remarkably, in the last years, we started to gather strong evidence showing that Plk1 might also play as a tumor suppressor. Two experimental studies correlate the reduced levels of Plk1 with cancer appearance due to the induction of aneuploidy and CIN [25,28]. The fact that reduced expression levels of a mitotic gene lead to tumor appearance is a common issue for many other mitotic regulators such as AurKA, AurKB, PTTG1, and Mad2. Nevertheless, genetic overexpression of the same mitotic genes also leads to tumor appearance, due to CIN events [69]. This feature demonstrates that a balanced expression level of certain mitotic genes is important to avoid CIN, ultimately leading to cancer appearance. In order to determine if Plk1 can indeed play as an oncogene, some laboratories performed Plk1 overexpression knock-in mouse models (Plk1-KI). Surprisingly, when these Plk1-KI mouse models were tested, we found that Plk1 elevated levels do not produce tumors, neither in vitro nor in vivo [27,65], indicating that Plk1 might not be an oncogene by itself. Indeed, overexpression of Plk1 lead to high levels of CIN and this is detrimental for tumor formation when combined with other oncogenes. Conversely, if Plk1 overexpression is combined with ionizing radiation, then mice harbor more tumors [65]. In summary, all Plk1 mouse models (either a gain or loss of function) ultimately lead to aneuploidy and CIN. In certain cases, this feature is tumor-prone [25,65], whereas in other circumstances, it leads to tumor suppression [27,28]. Although CIN has been typically considered to be a tumor-promoting mechanism [70], the effects of CIN on tumor development often depend on the cancer type [71]. Thus, we can find cases where CIN synergizes with tumor progression and on the contrary, high levels of CIN have been associated with improved prognosis [72,73]. It is very well known that Plk1 is intimately involved in chromosome segregation, and any alteration in Plk1 levels or activity can alter the balance of chromosome segregation, leading to CIN. The fact that this CIN derived from Plk1 activity alterations can promote or stop cancer progression demonstrates that the role of Plk1 in tumorigenesis is much more complex to understand, and deserves further attention.

## 6. Future Perspectives

When, how and why Plk1 can play as a tumor suppressor or can collaborate with tumor progression are still open questions with clinical relevance. PLK1 inhibition is an attractive anti-cancer strategy and some drugs that inhibit PLK1 kinase activity are currently in advanced clinical trials (Volasertib and Rigosertib) [23]. The fact that reduced activity of PLK1 can increase tumor development (in case of APC colorectal derived tumors [28]) suggests the need to re-evaluate the PLK1 inhibition strategy and try to define which tumors will benefit from this approach, and which ones will not. Further efforts in looking for the molecular mechanisms by which Plk1 plays as a tumor suppressor are needed. Moreover, looking for tumoral biomarkers that define a good or bad response to PLK1 inhibition is also needed to shed light on when Plk1 can be used as an anti-cancer strategy.

## Figures and Tables

**Figure 1 genes-10-00208-f001:**
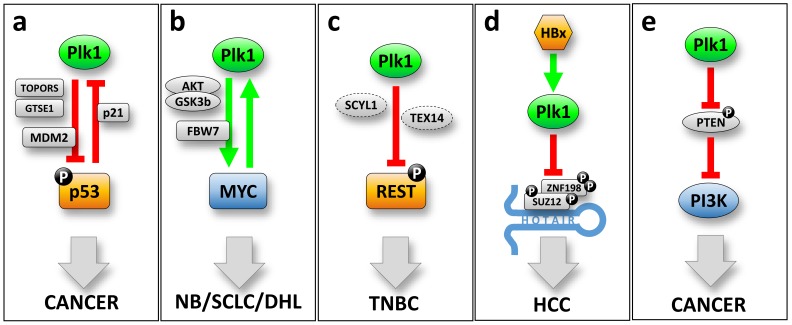
Proposed oncogenic mechanisms for Plk1. (**a**) The Plk1-p53 negative feedback loop. Plk1 leads to p53 destabilization, directly by phosphorylation of p53 or MDM2, or indirectly mediated by Topors and/or GTSE1. Transcriptionally, p53 is able to repress Plk1 expression directly binding to its promoter region. (**b**) The Plk1-MYC positive feedback loop. Plk1 can stabilize and increase the levels of the oncogenic transcription factor MYC, directly inhibiting the SCFFbw7 ubiquitin ligase complex, or by activation of the AKT-GSK3β kinase circuit. These mechanisms have been described for neuroblastoma (NB), small cell lung carcinoma (SCLC) and double hit lymphoma (DHL). (**c**) Plk1 overexpression leads to degradation of the tumor suppressor REST in triple negative breast cancer (TNBC). (**d**) Plk1 activation by the hepatitis B virus HBx factor leads to degradation of the transcription repression factors SUZ12 and ZNF198, both associated with the lncRNA HOTAIR in hepatocellular carcinoma. (**e**) Plk1 can activate the oncogenic PI3K pathway by inhibiting the PTEN tumor suppressor phosphatase. This leads to an increased cell proliferation rate and an elevated Warburg effect.

**Figure 2 genes-10-00208-f002:**
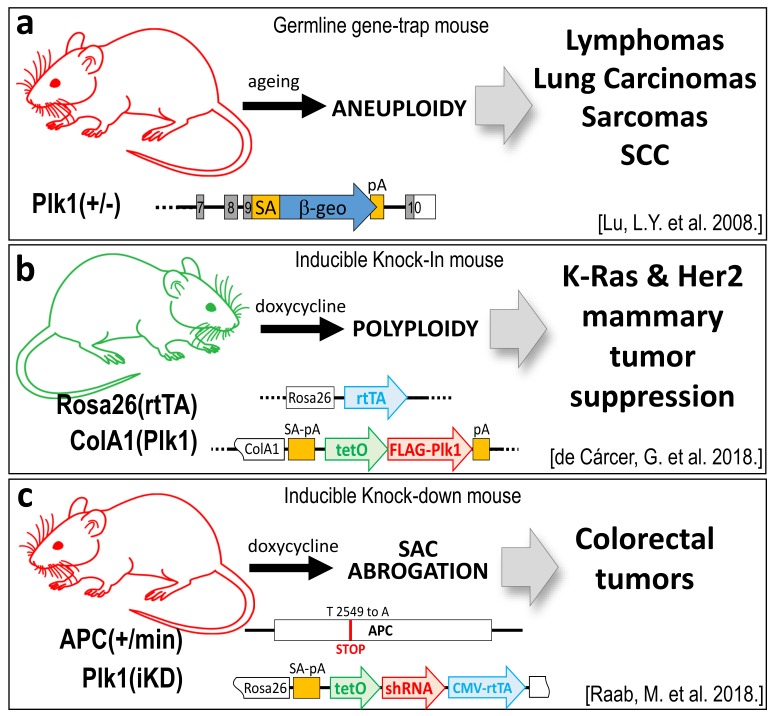
Plk1 genetic mouse models with tumor suppressor properties. (**a**) A germline heterozygous *Plk1* mouse (*Plk1+/**−*), generated by gene-trapping the exon 9 of the murine Plk1 gene, develops mainly lymphomas in the long latency, due to aneuploidy. (**b**) Inducible Plk1 knock-in (Plk1-KI), generated by inserting the human Plk1 cDNA down-stream of the tetO sequences in the endogenous ColA1 locus, and later combined with the Rosa26-rtTA transactivator. Plk1 overexpression leads to polyploidy and this stops mammary tumor generation by the Ras and her2 oncogenes. (**c**) *Plk1* inducible know-down mouse (*Plk1*-KD) is generated by inserting an shRNA under the tetO promoter u the Rosa26 locus. When combined with the colorectal model APC^min^ mice strain, downregulation of *Plk1* leads to an increase in colorectal papillomas due to Spindle Assembly Checkpoint (SAC) abrogation and chromosome instability (CIN) induction. (Plk1 loss of function models are represented in red colored mice. The Plk1 gain of function model is represented by a green mouse).

**Figure 3 genes-10-00208-f003:**
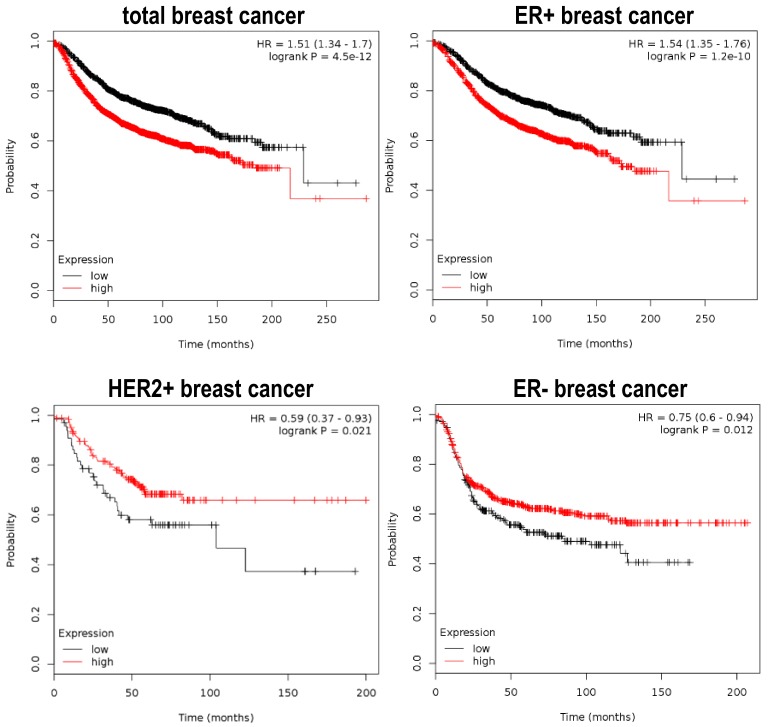
Differential prognostic value of Plk1 expression in breast tumor cancer. Plots depict the Relapse-Free Survival (RFS) ratio of different subtypes of breast cancer patients. Data is retrieved from the Kaplan Meier plotter initiative (www.kmplot.com), showing that Plk1 overexpressing tumors (red line) have a poor outcome in the estrogen ER+ subtype (n = 3951 samples) and total breast cancer tumors (n = 3951 samples). Conversely, ER− (n = 869 samples) and Her2+ (n = 252 samples) breast cancer patients, that express higher levels of Plk1 (red line), have better RFS ratios when compared to negative Plk1 tumors (black line).

**Figure 4 genes-10-00208-f004:**
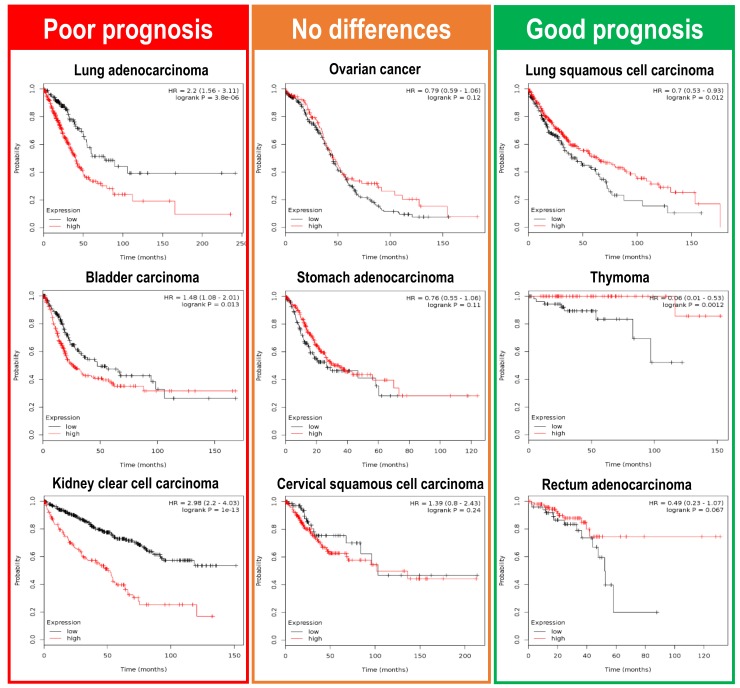
Clinical consequences of Plk1 upregulation in a pan-cancer RNAseq analysis. Overall survival (OS) analysis, in a large collection of different cancer types, retrieved from the Kaplan Meier plotter initiative (www.kmplot.com). mRNA sequence data obtained from nine different tumor types were plotted according to Plk1 expression levels (red line—high Plk1; black line—low Plk1). Lung, bladder, and kidney clear cell carcinomas have a poor OS when Plk1 levels are high. On the other hand, lung squamous cell carcinomas, thymoma, and rectum adenocarcinomas have a favorable OS outcome when Plk1 levels are elevated. Interestingly, there are also a variety of tumors where Plk1 expression have no influence in the OS for the patients (ovarian, stomach and cervical squamous carcinomas).

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
