# Peer review of "The Mitotic Cancer Target Polo-Like Kinase 1: Oncogene or Tumor Suppressor?"

_genes, 2019, doi:10.3390/genes10030208_

Round 1

Reviewer 1 Report

This is a nice comprehensive review onPLK1 and its oncologic implications, focusing on its tumor suppression function and the historical controversies around this issue. The review is updated and rather complete and has nice illustrations to guide the reader.  It also adds new information on patient survival / PLK1 expression and prognosis obtained from the Kaplan Meier plotter initiative.

I would suggest reviewing for minor writing issues like: spell check, definition of acronyms when they first appear (ex. CIN on line 222) and italicizing in vivo/in vitro…

It would also be adequate to indicate in figure 2 the references (authors, year) in each panel.

Although not absolutely necessary in the context of the discussion the author chose, it would be of interest to add PLK1 expression/survival data for triple negative breast cancer.

Author Response

Reviewer 1: Comments and Suggestions for Authors

This is a nice comprehensive review onPLK1 and its oncologic implications, focusing on its tumor suppression function and the historical controversies around this issue. The review is updated and rather complete and has nice illustrations to guide the reader.  It also adds new information on patient survival / PLK1 expression and prognosis obtained from the Kaplan Meier plotter initiative.

 >> I am very thankful to reviewer 1 evaluation and positive comments.

 I would suggest reviewing for minor writing issues like: spell check, definition of acronyms when they first appear (ex. CIN on line 222) and italicizing in vivo/in vitro…

 >> Regarding the minor writing issues, I apologize for these mistakes. As suggested I revised spelling and English grammar and correct all them.

 It would also be adequate to indicate in figure 2 the references (authors, year) in each panel.

 >> This is a nice suggestion. I upgrade figure 2 indicating the corresponding references for each panel.

 Although not absolutely necessary in the context of the discussion the author chose, it would be of interest to add PLK1 expression/survival data for triple negative breast cancer.

 >> Reviewer 1 makes a good point here. I also was thinking on this possibility during the manuscript writing. The Plk1 triple negative breast cancer (TNBC) survival plot shows no differences in survival. This is already shown in the supplementary figure 8 of de Carcer et al., 2018 (ref [27]). To avoid any copyright conflict with reference [28], I decide to slightly change this figure by showing relapse free survival (RFS) instead of overall survival (OS), and exchange the TNBC plot by the total BC plot. 

Reviewer 2 Report

Perfect review, congratulation! 

There are plenty of studies published on PLK1 role in the cell mitosis and cytokinesis as well as its aberrant expression in  different cancer cells and its avail as anti-cancer  therapy target. What is really missing – the critical view on those  studies. Since 2013 Cholewa BD, Liu X and Ahmad N paper in Cancer  Research  “The role of polo-like kinase 1 in carcinogenesis: cause or consequence? “ where the authors made careful conclusion –  “On  the basis  of the available literature, it may be somewhat premature to draw a  definitive conclusion on the role of Plk1 in carcinogenesis” –  no one PLK1 review  paper tried to look at flip sides of the same coin.

So,  the Guillermo de Cárcer manuscript has appeared just in time with  critical and careful analysis of all pros and cons about PLK1 role as  oncogene.   The  author analyzed purposefully  almost all main publications dedicated PKL1 and made reasoned  conclusions that is why I recommend to publish this work. This fresh  view on PLK1 functions in cancer cell  will be useful for researches working in this field.

Author Response

Reviewer 2: Comments and Suggestions for Authors

Perfect review, congratulation! There are plenty of studies published on PLK1 role in the cell mitosis and cytokinesis as well as its aberrant expression in different cancer cells and its avail as anti-cancer therapy target. What is really missing – the critical view on those studies. Since 2013 Cholewa BD, Liu X and Ahmad N paper in Cancer Research “The role of polo-like kinase 1 in carcinogenesis: cause or consequence?” where the authors made careful conclusion –  “On  the basis  of the available literature, it may be somewhat premature to draw a  definitive conclusion on the role of Plk1 in carcinogenesis” –  no one PLK1 review  paper tried to look at flip sides of the same coin.

So, the Guillermo de Cárcer manuscript has appeared just in time with critical and careful analysis of all pros and cons about PLK1 role as oncogene. The author analyzed purposefully almost all main publications dedicated PLK1 and made reasoned conclusions that is why I recommend to publish this work. This fresh view on PLK1 functions in cancer cell will be useful for researches working in this field.

 >> I am very glad to read such a positive evaluation. Thank you so much!